# Participatory Personalization in Classification

**Hailey Joren** [1]  **Chirag Nagpal** [2]  **Katherine Heller** [3]  **Berk Ustun** [1]

## Abstract

Machine learning models are often personalized with information that is protected, sensitive, self-reported, or costly to acquire. These models use information about people, but do not facilitate nor inform their *consent*. Individuals cannot opt out of reporting personal information to a model, nor tell if they benefit from personalization in the first place. We introduce a family of classification models, called *participatory systems*, that let individuals opt into personalization at prediction time. We present a model-agnostic algorithm to learn participatory systems for personalization with categorical group attributes. We conduct a comprehensive empirical study of participatory systems in clinical prediction tasks, benchmarking them with common approaches for personalization and imputation. Our results demonstrate that participatory systems can facilitate and inform consent while improving performance and data use across all groups who report personal data.

## 1. Introduction

Machine learning models routinely assign predictions to *people* – be it to screen a patient for a mental illness [33], their risk of mortality in an ICU [43], or their likelihood of responding to treatment [1]. Many models in such applications use personal information to target heterogeneous subpopulations. Typically, models are *personalized* with categorical attributes that define groups [i.e., "categorization" as per 25]. In medicine, for example, clinical prediction models use *group attributes* that are *protected* (e.g., sex in the CHA$_2$DS$_2$ Score for Stroke Risk), *sensitive* (e.g., HIV status in the VA COVID-19 Mortality Score), *self-reported* (e.g., age_menarche in the Gail Breast Cancer Risk Score), or *costly* to acquire (e.g., leukocytosis in the Alvarado Appendicitis Score).

[1]University of California San Diego [2]Carnegie Mellon University [3]Google. Correspondence to: Hailey Joren <hjoren@ucsd.edu>.

*Workshop on Interpretable ML in Healthcare at International Conference on Machine Learning (ICML)*, Honolulu, Hawaii, USA.

Online platforms that solicit personal data are designed to support *informed consent*: individuals can opt out of providing personal data and understand how it will be used [see, e.g., personal data guidelines in GDPR, OECD privacy guidelines 24, 39]. Personalized models do not provide such functionality: individuals cannot opt out of reporting data used to personalize their predictions, nor tell if it would improve their predictions. Practitioners assume that data available for training will be available at inference time. In practice, this assumption has led to a proliferation of models that use information that individuals may be unwilling or unable to report at prediction time [see e.g., the Denver HIV Risk Score 27, which asks patients to report age, gender, race, and sexual_practices]. In tasks where individuals self-report, they may not voluntarily report information that could improve their predictions or may report information that is incorrect.

The need to inform consent in machine learning stems from the fact personalization may not improve performance for each group that reports personal data [51]. In practice, a personalized model can perform *worse* or the same as a *generic model* fit without personal information for a group with specific characteristics. Such models violate the implicit promise of personalization as individuals report personal information without receiving a tailored performance gain in return. These instances of *worsenalization* are prevalent, hard to detect, and hard to avoid [see 51, 41] – but would be resolved by allowing individuals to opt out of personalization and disclosing its expected gains (see Fig. 1).

In this work, we introduce a family of classification models that operationalize informed consent called *participatory systems*. Participatory systems *facilitate consent* by allowing individuals to report personal information at prediction time. Moreover, they *inform consent* by showing how reporting personal information will change their predictions. Models that facilitate consent operate as *markets* in which individuals trade personal information for performance gains. This work seeks to develop systems that: (i) perform as well as possible when individuals opt in (to incentivize voluntary reporting) or opt out (to safeguard against abstention); (ii) provide opportunities for individuals to make informed decisions about data provision. The resulting systems can lead

| Group | Data | | Personalized | | Generic | | Traditional Personalization
groups receive predictions from $h$ | | | Minimal Participatory System
groups opt into predictions from $h$ or $h_0$ | | |
|---|---|---|---|---|---|---|---|---|---|---|---|---|
| | | | | | | | Model | Data Use | Gain | Model | Data Use | Gain |
| $g$ | $n_g^+$ | $n_g^-$ | $h$ | $R_g(h)$ | $h_0$ | $R_g(h_0)$ | | $r$ | $\Delta R_g(h, h_0)$ | | $r$ | $\Delta R_g(h, h_0)$ |
| female, old | 0 | 24 | + | 24 | − | 0 | $h$ | female, old | −24 | $h_0$ | ∅ | 0 |
| female, young | 25 | 0 | + | 0 | − | 25 | $h$ | female, young | 25 | $h$ | female, young | 25 |
| male, old | 25 | 0 | + | 0 | − | 25 | $h$ | male, old | 25 | $h$ | female, young | 25 |
| male, young | 0 | 27 | − | 0 | − | 0 | $h$ | male, young | 0 | $h_0$ | ∅ | 0 |
| **Total** | 50 | 51 | | 24 | | 50 | | | 26 | | | 50 |

**Figure 1:** Simple classification task where participation improves performance and limits data use. The best generic linear model $h_0 : \mathcal{X} \times \mathcal{Y}$ fits a decision boundary that predicts the majority class (−) for all samples, as there are $n^+ = 50$ positive and $n^- = 51$ negative examples. The best personalized linear model with a one-hot encoding of group attributes $h : \mathcal{X} \times \mathcal{G} \to \mathcal{Y}$ fits a decision boundary using $\mathcal{G} = \texttt{sex} \times \texttt{age}$ that classifies the majority of points correctly ([female, young], [male, old] and [male, young]). Under traditional personalization, individuals must report group membership to $h$. Here, personalization with $h$ reduces total error from 50 to 24 ($\Delta R_g(h, h_0) = 26$) but assigns the same predictions as the generic model to [male, young] ($\Delta R_g(h, h_0) = 0$) and detrimental predictions to [female, old] ($\Delta R_g(h, h_0) = -24$). In a minimal participatory system, individuals can opt in or out of personalization to receive predictions from $h$ or $h_0$. Here, individuals in groups [female, old] and [male, young] would opt out of personalization, leading to an overall error of 0 ($\Delta R_g(h, h_0) = 50$) and a reduction in unnecessary data collection ($\varnothing$).

to large improvements in performance and data use across all groups who report personal data, maximizing the gains of personal information when it improves performance and limiting data collection when it does not.

**Related Work** Participatory systems support modern principles of responsible data use such as *informed consent* and *collection limitation* – i.e., data should be collected with the consent of a data subject and restricted to only what is necessary. These principles are articulated in, e.g., OECD privacy guidelines [39], the GDPR [24], and the California Consumer Privacy Act [16]. These principles stem from extensive work on the right to data privacy [31]. They are motivated, in part, by research showing that individuals care deeply about their ability to control personal data [11, 4, 9] and differ considerably in their desire or capacity to share it [see e.g. 10, 40, 17, 18, 8, 37, 6].

We consider models that are personalized with categorical attributes that encode personal characteristics. [i.e., "categorization" rather than "individualization" as per 25]. Modern techniques for learning with categorical attributes [see e.g., 2, 47] use them to improve performance at a population level – e.g., by accounting for higher-order interaction effects [14, 36, 57] or recursive partitioning [23, 15, 13, 12]. Our work provides methods to achieve these goals in settings where models use features that are optional or costly to acquire [see e.g., 7, 8, 60, 52].

Our work is broadly related to algorithmic fairness in that we seek to improve model performance at a group level. Recent work has demonstrated that personalization with group attributes does not uniformly improve performance and can result in less accurate predictions at a group level [see 51, 41, 56]. The proposed systems can mitigate this effect by allowing individuals to opt out of such instances of "worsenalization." This line of work complements research on

preference-based fairness [61, 56, 34, 58, 22], on ensuring fairness across complex group structures [32, 28, 26], and on the study of privacy across subpopulations [10, 53].

## 2. Participatory Systems

We consider a classification task where we personalize a model with categorical attributes. We start with a dataset $\{(\boldsymbol{x}_i, y_i, \boldsymbol{g}_i)\}_{i=1}^n$ where each example consists of a feature vector $\boldsymbol{x}_i \in \mathbb{R}^d$, a label $y_i \in \mathcal{Y}$, and a vector of $k$ categorical attributes $\boldsymbol{g}_i = [g_{i,1}, \ldots, g_{i,k}] \in \mathcal{G}_1 \times \ldots \times \mathcal{G}_k = \mathcal{G}$. We refer to $\mathcal{G}$ as *group attributes*, and to $\boldsymbol{g}_i$ as the *group membership* of person $i$. We let $n_{\boldsymbol{g}} := |\{i \,|\, \boldsymbol{g}_i = \boldsymbol{g}\}|$ denote the size of group $\boldsymbol{g}$, and $|\mathcal{G}_i|$ denote the number of categories for group attribute $i$.

We use the dataset to fit a model $h : \mathcal{X} \times \mathcal{G} \to \mathcal{Y}$ by empirical risk minimization with a loss $\ell : \mathcal{Y} \times \mathcal{Y} \to \mathbb{R}_+$. Given a model $h$, we denote its empirical risk and true risk as $\hat{R}(h)$ and $R(h)$. We denote the true risk and empirical risk of $h$ for group $\boldsymbol{g} \in \mathcal{G}$ when they are assigned the personalized predictions for group $\boldsymbol{g}' \in \mathcal{G}$ as:

$$R_{\boldsymbol{g}}(h(\cdot, \boldsymbol{g}')) := \mathbb{E}\left[\ell\left(h_{\boldsymbol{x}, \boldsymbol{r}'}, y\right) \mid \mathcal{G} = \boldsymbol{g}'\right]$$
$$\hat{R}_{\boldsymbol{g}}(h(\cdot, \boldsymbol{g}')) := \frac{1}{n_{\boldsymbol{g}}} \sum_{i : \boldsymbol{r}_i = \boldsymbol{g}} \ell\left(h_{\boldsymbol{x}_i, \boldsymbol{g}'}, y_i\right).$$

We assume that groups prefer more accurate predictions. Given models $h$ and $h'$, users in group $\boldsymbol{g}$ prefer $h$ when $R_{\boldsymbol{g}}(h) < R_{\boldsymbol{g}}(h')$. This assumption holds in settings where individuals prefer more accurate predictions – e.g., when predicting the risk of an illness [see e.g., 49, 35, 48] or recommending content or products on online platforms [38]. It does not hold in applications where some individuals prefer inaccurate predictions – e.g., "polar" clinical prediction tasks such as predicting the risk of organ failure for a transplant [42].

**Facilitating Consent** We consider models where individuals consent to personalization by reporting group attributes at prediction time. We let $\varnothing$ denote an attribute that was not reported, and let $\boldsymbol{r}_i = [r_{i,1}, \ldots, r_{i,k}] \in \mathcal{R} \subseteq \mathcal{G} \times \varnothing$ denote the *reported group membership* of person $i$. For example, a person with $\boldsymbol{g}_i = [\texttt{female}, \texttt{HIV} = +]$ would have $\boldsymbol{r}_i = [\texttt{female}, \varnothing]$ if they only report $\texttt{sex}$, and $\boldsymbol{r}_i = \boldsymbol{\varnothing} := [\varnothing, \ldots, \varnothing]$ if they opt out of personalization entirely.

We associate models with a set of *reporting options* $\mathcal{R}$. A traditional model where each person must report group attributes has $\mathcal{R} = \mathcal{G}$. A model where each person could report any subset of group attributes has $\mathcal{R} = \mathcal{G} \times \boldsymbol{\varnothing}$. The truthful reporting options of a group $\boldsymbol{g}$ include all $\boldsymbol{r} \in \dim \boldsymbol{g} := \boldsymbol{g} \times \boldsymbol{\varnothing}$. We use $n_{\boldsymbol{r}}$ to denote the number of individuals who could truthfully report $\boldsymbol{r} \in \mathcal{R}$. We represent individual decisions to opt into personalization at prediction time through a *reporting interface*, defined below.

**Definition 1** (Reporting Interface)**.** Given a personalized classification task with group attributes $\mathcal{G}$, a *reporting interface* is a tree $T$ whose nodes represent attributes reported at prediction time $\mathrm{nodes}(T) = \mathcal{R} \subseteq \mathcal{G} \times \boldsymbol{\varnothing}$. The tree is rooted at $\mathrm{root}(T) = \boldsymbol{\varnothing}$ and branches as an individual reports group attributes. Given a node $\boldsymbol{r}$, we denote its child $\boldsymbol{r}'$ with $\boldsymbol{r} = \mathrm{pa}(\boldsymbol{r}')$. Each parent-child pair maps to a *reporting decision*, and the height of the tree represents the number of reporting decisions. If users can opt out of personalization $\boldsymbol{\varnothing} \in \mathrm{nodes}(T)$ then the tree supports *consent* to personalization.

**Definition 2.** Given a personalized classification task with group attributes $\mathcal{G}$, a *participatory system* with reporting interface $T$ is a model $f_T : \mathcal{X} \times \mathcal{R} \to \mathcal{Y}$ that obeys:

I. *Incentive Compatibility*: Opting into personalization should improve performance

$$R_{\boldsymbol{r}}(f_T(\cdot, \boldsymbol{r})) < R_{\boldsymbol{r}}(f_T(\cdot, \boldsymbol{r}'))$$
$$\forall \boldsymbol{r}, \boldsymbol{r}' \text{ such that } \boldsymbol{r}' = \mathrm{pa}(\boldsymbol{r})$$

II. *Baseline Performance*: Opting out of personalization should guarantee the performance of a *generic model* $h_0 : \mathcal{X} \to \mathcal{Y}$ where $h_0 \in \mathrm{argmin}_{h \in \mathcal{H}} R(h)$.

$$R_{\boldsymbol{g}}(f_T(\cdot, \boldsymbol{\varnothing})) \leq R_{\boldsymbol{g}}(h_0(\cdot)) \text{ for all groups } \boldsymbol{g} \in \mathcal{G}.$$

We denote the gains of personalization for group $\boldsymbol{g}$ in terms of true risk and empirical risk as $\Delta_{\boldsymbol{g}}(\boldsymbol{r}, \boldsymbol{\varnothing}) := R_{\boldsymbol{g}}(f_T(\cdot, \boldsymbol{\varnothing})) - R_{\boldsymbol{g}}(f_T(\cdot, \boldsymbol{r}))$ and $\hat{\Delta}_{\boldsymbol{g}}(\boldsymbol{r}, \boldsymbol{\varnothing}) := \hat{R}_{\boldsymbol{g}}(f_T(\cdot, \boldsymbol{\varnothing})) - \hat{R}_{\boldsymbol{g}}(f_T(\cdot, \boldsymbol{r}))$.

Here, the *Incentive Compatibility* property guarantees that opting into personalization will indeed improve the model's performance. It ensures that when individuals provide their personal information, the system can effectively leverage that data to deliver more accurate predictions. Conversely, the *Baseline Performance* property serves as a safeguard for those who opt out of personalization. It assures individuals that without sharing their personal information, they will still receive the performance of a generic model trained on a dataset without group attributes. Together, these properties uphold the principle of rationality, in which users always receive gains from personalization such that $\Delta_{\boldsymbol{g}}(\boldsymbol{r}, \boldsymbol{\varnothing}) > 0$ for all $\boldsymbol{g} \in \mathcal{G}, \boldsymbol{r} \in \dim \boldsymbol{g}$. By balancing personalization incentives with respect for user preferences, these properties promote a fair and rational approach to the use of personal information while adhering to the principles of data minimization.

The simplest way to facilitate consent is to impute the group membership of individuals who opt out of personalization at prediction time. Imputation allows individuals to opt out of personalization, but does not provide guarantees on the accuracy of predictions for individuals who opt out. Formally, imputation violates the baseline performance guarantee of a participatory system. That is, if users opt out of personalization ($\boldsymbol{r} = \boldsymbol{\varnothing}$), they may receive a less accurate prediction than they would have from a generic model $R_{\boldsymbol{g}}(f_T(\cdot, \hat{\boldsymbol{g}})) > R_{\boldsymbol{g}}(h_0(\cdot))$ where $\hat{\boldsymbol{g}}$ denote the imputed attributes. In a setting where we could perfectly impute group membership, for example, a group might may be assigned better predictions from a generic model (see Fig. 1). In the worst case, imputation may be incorrect, leading to predictions that are even more inaccurate than those of the generic or personalized model.

## 3. Learning Participatory Systems

This section describes a model-agnostic algorithm to learn participatory systems that ensure incentive compatibility and baseline performance in deployment.

We outline our procedure in Algorithm 1 to learn the three kinds of participatory systems in Fig. 2. The procedure takes as input a pool of candidate models $\mathcal{M}$ and datasets for assignment and pruning $\mathcal{D}^{\mathrm{train}}$ and $\mathcal{D}^{\mathrm{valid}}$. It outputs a collection of participatory systems that ensures the desiderata of participatory systems on test data. The procedure combines three routines to: (1) generate viable reporting interfaces (Line 1); (2) assign models over the interface (Line 3); (3) prune the system to limit unnecessary data collection (Line 4). We present a complete procedure for each routine Appendix B, and discuss them below.

**Model Pool** Our procedure takes as input a *pool of personalized models* $\mathcal{M}$ to assign over a reporting interface. At a minimum, a pool should contain a personalized model $h$ for individuals who opt into personalization, and a generic

| Interface | Sample System | Description |
|---|---|---|

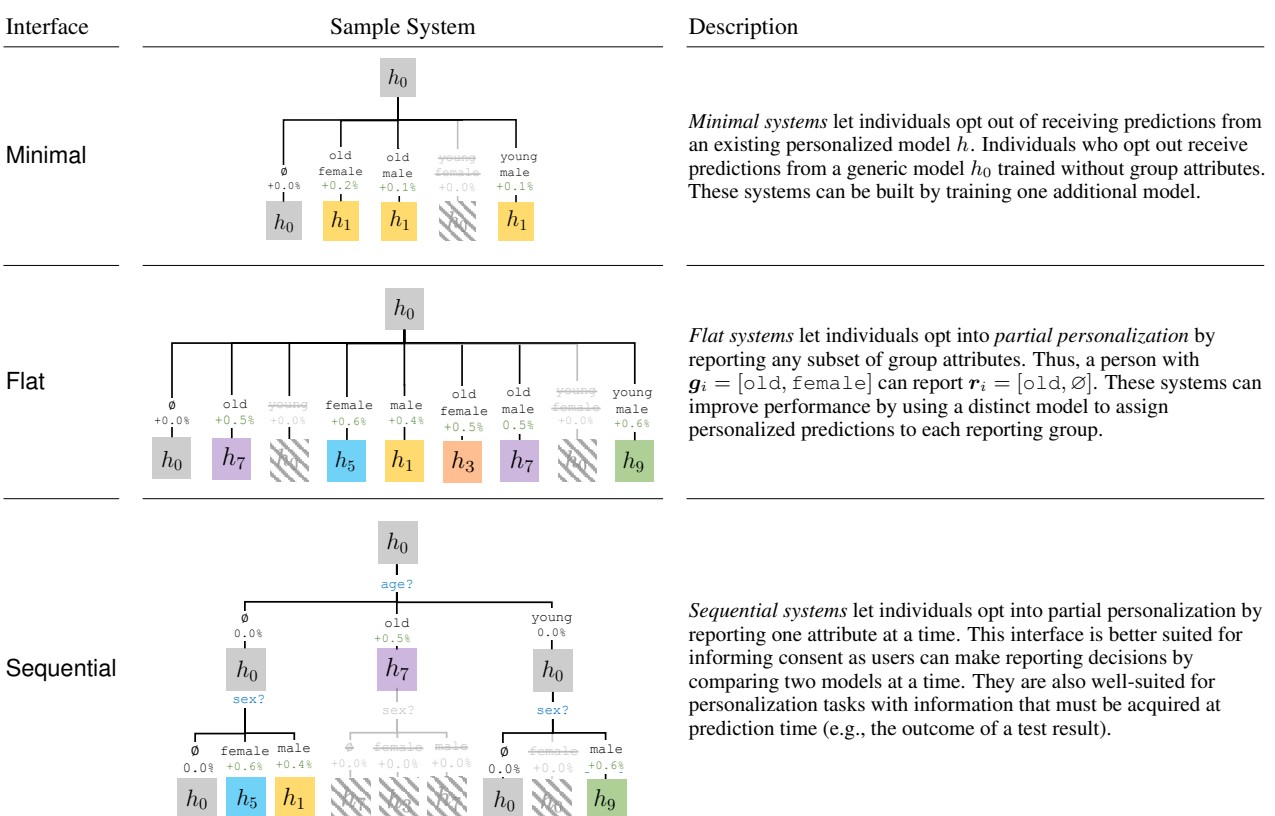

**Figure 2:** Participatory systems for personalized classification task with group attributes $\texttt{sex} \times \texttt{age} = [\texttt{male}, \texttt{female}] \times [\texttt{old}, \texttt{young}]$. Each system allows a person to opt out of personalization by reporting $\varnothing$, and informs this choice through the gains of personalization (e.g., +0.2% gain in accuracy). Systems minimize data use by removing reporting options that do not lead to gain (e.g., $[\texttt{young}, \texttt{female}]$ is pruned in all systems as it leads to a gain $\leq 0.0\%$). We show reporting options pruned using grey-striped boxes.

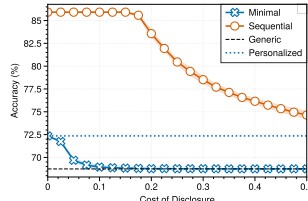

**Figure 3:** Performance profile of participatory systems for the saps dataset for individuals in group $\boldsymbol{g}_i = [\texttt{30+, HIV=+}]$. We plot out-of-sample performance for different levels of participation in the target population. We control participation by varying the cost of reporting in a simulated model of individual disclosure. As shown, minimal and sequential systems always outperform the generic model regardless of participation. In regimes where the cost of reporting is low, participation is high. Consequently, a minimal system will achieve the same performance as a personalized model, and a sequential system to achieve the performance of the component model for this subpopulation. We provide details and results in Appendix A.

---

**Algorithm 1** Learning Participatory Systems

Input: $\mathcal{M} : \{h : \mathcal{X} \times \mathcal{G} \to \mathcal{Y}\}$      *pool of candidate models*

Input: $\mathcal{D}^{\text{assign}} = \{(\boldsymbol{x}_i, \boldsymbol{g}_i, y_i)\}_{i=1}^{n^{\text{assign}}}$      *training dataset*

Input: $\mathcal{D}^{\text{prune}} = \{(\boldsymbol{x}_i, \boldsymbol{g}_i, y_i)\}_{i=1}^{n^{\text{prune}}}$      *validation dataset*

1: $\mathbb{T} \leftarrow \textsf{ViableTrees}(\mathcal{G}, \mathcal{D}^{\text{assign}})$    $|\mathbb{T}| = 1$ *for minimal & flat systems*

2: **for** $T \in \mathbb{T}$ **do**

3:      $T \leftarrow \textsf{AssignModels}(T, \mathcal{M}, \mathcal{D}^{\text{assign}})$      *assign models*

4:      $T \leftarrow \textsf{PruneLeaves}(T, \mathcal{D}^{\text{prune}})$      *prune models*

5: **end for**

**Output** $\mathbb{T}$, collection of participatory systems

---

trade-offs between groups. Using a pool safeguards against these effects by drawing on models from model classes that have been personalized using different techniques for each reporting group. By default, we include a model for each reporting group trained using only data for that group, as such models can perform well on heterogeneous subgroups [44, 56, 51].

**Enumerating Reporting Interfaces** We call the ViableTrees routine in Line 1 to enumerate *viable* reporting interfaces. We only call this routine for sequential systems

---

model $h_0$ for individuals who opt out of personalization. A single personalized model can perform unreliably across reporting groups due to differences in the data distribution or

since $\mathbb{T}$ consists of a single tree known a priori for minimal and flat systems. ViableTrees takes as input a group attributes $\mathcal{G}$ and a dataset $\mathcal{D}^{\text{assign}}$ and returns all $m$-ary trees that obey constraints on sample size and reporting (e.g., users who report `male` should report `age` before `HIV`). By default, we only generate trees so that we have sufficient data to estimate gains at each node of the reporting interface (e.g., trees whose leaves contain at least one positive sample, one negative sample, and $n_{\boldsymbol{r}} \geq d + 1$ samples to avoid overfitting). In general, ViableTrees scales to tasks with $\leq 8$ group attributes. Beyond this limit, one can reduce the size of the enumeration by specifying ordering constraints or a threshold number of trees to enumerate before stopping. For a task with 3 binary group attributes $\mathbb{T}$ contains 24 3-ary trees of depth 3. Given a complete ordering of all 3 group attributes, $\mathbb{T}$ would contain 1 tree. We can also consider a greedy algorithm (see Appendix B.4), which may be practical for large-scale problems at the expense of posthoc flexibility and full search space exploration.

**Model Assignment**    We assign each reporting group a model using the AssignModels routine in Line 3. Given a reporting group $\boldsymbol{r}$, we consider all models that could use group membership $\boldsymbol{r}_i \in \dim \boldsymbol{r}$. Thus, a group that reports `age` and `sex` could be assigned predictions from a model that requires `age`, `sex`, both, or neither. This implies that we can always assign the generic model to any reporting group, ensuring that the model at each node performs as well as the generic model on out-of-sample data (i.e., *baseline performance* in Definition 2).

**Pruning**    ViableTrees may output trees that violate incentive compatibility by requesting personal information that does not reliably improve performance. This can happen when a system assigns a model that performs equally well to nested reporting groups – see, e.g., Fig. 2 where the Flat system assigns $h_0$ to $[\texttt{female}, \varnothing]$ and $[\texttt{female}, \texttt{young}]$. The Prune routine in Line 4. This routine takes as input a participatory system $f_T$ and a validation sample $\mathcal{D}^{\text{prune}}$ and outputs a system $f_{T'}$ with a pruned interface $T' \subseteq T$ that will not request data in such cases through a bottom-up pruning procedure. The procedure calls a one-sided hypothesis test at each node:

$$H_0 : \Delta_{\boldsymbol{r}}(\boldsymbol{r}, \text{pa}(\boldsymbol{r})) \leq 0 \qquad H_A : \Delta_{\boldsymbol{r}}(\boldsymbol{r}, \text{pa}(\boldsymbol{r})) > 0$$

The test checks if each reporting group $\boldsymbol{r}$ receives more accurate predictions using the personalized model assigned by the current interface $T$ or its parent $f_T(\cdot, \text{pa}(\boldsymbol{r}))$. Here, $H_0$ assumes a reporting group prefers the parent model. Thus, we reject $H_0$ when evidence suggests that $f_T(\cdot, \boldsymbol{r})$ performs better for $\boldsymbol{r}$ on pruning data. The exact test should be chosen based on the performance metric for the underlying prediction task. In general, we can use a bootstrap hypothesis test [20], and draw on more powerful tests for salient performance metrics [e.g., 21, 19, 50, for accuracy and AUC].

**On Computation**    Our approach provides practitioners with several options to learn participatory systems. For example, one can train only two models and build a minimal system, or train a flat or sequential system with a limited number of models in the pool. Nevertheless, the primary bottleneck when learning participatory systems is *data* rather than *compute*. Given a finite sample dataset, we are limited in the number of categorical attributes used for personalization. This is because we require a minimum number of samples for each intersectional group to train a personalized model and evaluate its performance. Given that the number of intersectional groups increases exponentially with each attribute, we quickly enter a regime where we cannot train models for a given group (e.g., because we lack sufficient labels) or reliably evaluate its gain for assignment and pruning [see 41].

## 4. Experiments

We benchmark participatory systems on real-world clinical prediction tasks. Our goal is to evaluate these approaches in terms of performance, data usage, and consent in applications where individuals have a low cost of reporting. We include code to reproduce these results in an anonymized repository.

### 4.1. Setup

We consider six classification tasks for clinical decision support where we personalize a model with group attributes that are protected or sensitive (see Table 2 and Appendix C). Each task pertains to an application where the information used for personalization is readily available, relevant to the prediction task, and likely to be disclosed given laws surrounding the privacy and confidentiality of health data [54]. Given these conditions, we expect individuals to have a low cost of reporting – and therefore report personal information so long as there is any benefit [11, 5]. One exception is `cardio_eicu` and `cardio_mimic`, which are personalized based on race and ethnicity. We note that the use of race in clinical risk scores should be approached with caution [59]; participatory systems offer one way to safeguard against inappropriate use.

We split each dataset into a test sample (20%, to evaluate out-of-sample performance) and a training sample (80% for training, pruning, assignment, and estimating gains to show users). We train three kinds of personalized models for each dataset:

- *Static*: These are models are personalized using a one-hot encoding of group attributes (1Hot), and a one-hot

encoding of intersectional groups (mHot)

- *Imputed*: These are variants of static models where we impute the values of group attributes (KNN-1Hot, KNN-mHot). In practice, the performance of imputation will range between the values reported for 1Hot and KNN-mHot (100% opt-in) and those reported for KNN-1Hot, KNN-mHot(100% opt-out).

- *Participatory*: These are participatory systems built using our approach. These include: Minimal, a minimal system built from 1Hot and its generic counterpart; and Flat and Seq, flat and sequential systems built from 1Hot, mHot and their generic counterparts.

We train all models – personalized models and the components of participatory systems – from the same model class, and evaluate them using the metrics in Table 1. We repeat the experiments four times, varying the model class (logistic regression, random forests) and the salient performance metric (error rate for decision-making and AUC for ranking) to evaluate the sensitivity of our findings with respect to model classes and classification taskss.

### 4.2. Results

We show results for logistic regression models and error rate in Table 2, and for other model classes and classification tasks in Appendix D. In what follows, we discuss these results.

**On Performance**   Our results in Table 2 show that participatory systems can lead to performance improvements across reporting groups. Here, Flat and Seq achieve the best overall performance on 6/6 datasets and improve the gains from personalization for every reporting group on 5/6 datasets. In contrast, traditional models improve overall performance while reducing performance at a group level (see rationality violations on 5 datasets for 1Hot, mHot, KNN-1Hot, and KNN-mHot). The performance benefits from participatory systems stem from: (i) allowing users to opt out of these instances of "worsenalization" and (ii) assigning personalized predictions with multiple models. Using Table 2, we can measure the impact of (i) by comparing the performance of Minimal vs. 1Hot), and (ii) by comparing the performance of Flat or Seq vs. Minimal. For example, on apnea, 1Hot improves performance at a population level but exhibits a significant rationality violation for group [30_to_60,male], meaning they would have been better off with a generic model that did not use their personal data. By comparing the performance of 1Hot to Minimal, we see that allowing users to opt out of worsenalization reduces test error from 29.1% to 28.9%. By comparing the performance on Minimal to Flat and Seq, we see that using multiple models can further reduce test error from 28.9% to 24.1%.

**On Informed Consent**   Our results show how Flat and Seq systems can inform consent by allowing users to report a subset of group attributes (e.g., by including reporting options such as [30+,∅] or [∅,HIV+]). Although both Flat and Seq systems allow for partial personalization, they differ in their capacity to inform consent. In a flat system, users may inaccurately gauge the marginal benefit of reporting an attribute by comparing the gains between reporting options. For example, in Fig. 4, users who are HIV positive would see a gain of 3.7% for reporting [∅,HIV+], and 16.7% for reporting [30+,HIV+] and may mistakenly conclude that the gain of reporting age is 16.7% − 3.7% = 13.0%. This estimate incorrectly presumes that the gains of 3.7% were distributed equally across age groups. Sequential systems directly inform users of the gains for partial reporting. In the sequential system, group [30+,HIV+] is informed that they would see a marginal gain of 21.5% for reporting age, while group [<30,HIV+] is informed they would see a marginal gain of reporting age of 0.0%.

**On Data Use**   Our results show that participatory systems perform better across all groups while requesting less personal data on 6/6 datasets. For example, on cardio_eicu, Seq reduces error by 11.3% compared to 1Hot while requesting, on average, 83.3% of the data needed by 1Hot. In general, participatory systems can limit data use where personalization does not improve performance, e.g., on lungcancer. Even as attributes like sex or age may be readily reported by patients for any performance benefit, limiting data use is valuable when there is a tangible cost associated with data collection – e.g., when models make use of rating scale for a mental disorder that must be administered by a clinician [46]. The potential for data minimization varies substantially across prediction tasks. On apnea, for example, we can prune 6 reporting options when building a Seq for decision making (which optimizes error) but 4 options for Seq for ranking (which optimizes AUC; see Appendix D.1).

**On the Benefits of a Model-Agnostic Approach**   Our findings highlight some of the benefits of a model-agnostic approach, in which we can draw on a rich set of models to achieve better performance while mitigating harm. The resulting system can balance training costs with performance benefits. For example, on the cardio_eicu dataset, training a pool of 126 candidate models takes 6.5 seconds for logistic regression and 27.5 seconds for random forests on a single CPU. We can also ensure generalization across reporting groups – e.g., by a generic model fit from a complex model class, and personalized models fit from a simpler model class. As expected, fitting for a complex model class can lead to considerable changes in overall accuracy – e.g., we can reduce overall test error for a personalized model from 20.4% to 14.1% on saps by fitting a random forest

| Metric | Definition | Description |
|---|---|---|
| Overall Performance | $\sum_{\boldsymbol{g} \in \mathcal{G}} \frac{n_{\boldsymbol{g}}}{n} \hat{R}_{\boldsymbol{g}}(h_{\boldsymbol{g}})$ | Population-level performance of a personalized system/model, computed as a weighted average over all groups |
| Overall Gain | $\sum_{\boldsymbol{g} \in \mathcal{G}} \frac{n_{\boldsymbol{g}}}{n} \hat{\Delta}_{\boldsymbol{g}}(\boldsymbol{g}, \varnothing)$ | Population-level gain in performance of a personalized system/model over its generic counterpart |
| Group Gains | $\min_{\boldsymbol{g} \in \mathcal{G}} / \max_{\boldsymbol{g} \in \mathcal{G}} \hat{\Delta}_{\boldsymbol{g}}(\boldsymbol{g}, \varnothing)$ | Range of gains of a personalized system/model over its generic counterpart across all groups |
| Rationality Violations | $\sum_{\boldsymbol{g} \in \mathcal{G}} \mathbb{I}[\text{reject } H_0]$ | Number of rationality violations detected using a bootstrap hypothesis test with 100 resamples and significance level of 10% where $H_0 : \Delta_{\boldsymbol{g}}(\boldsymbol{g}, \varnothing) \geq 0$ and $H_A : \Delta_{\boldsymbol{g}}(\boldsymbol{g}, \varnothing) < 0$ |
| Imputation Risk | $\min_{\boldsymbol{g} \in \mathcal{G}} \hat{\Delta}_{\boldsymbol{g}}(\boldsymbol{g}, \boldsymbol{g}')$ | Risk to performance of imputation, or the worst possible performance to a group given they are imputed with the attributes of group $\boldsymbol{g}'$. Relevant for static models only |
| Options Pruned | $(|\mathcal{R}| - |\mathcal{R}(h)|)/|\mathcal{R}|$ | Number of reporting options pruned from a model or system. Here, $\mathcal{R}(h)$ denotes the options that are available after $h$ has been pruned while $\mathcal{R}$ denotes options available before pruning. |
| Data Use | $\sum_{\boldsymbol{g} \in \mathcal{G}} \frac{n_{\boldsymbol{g}}}{n} \frac{\text{requested}(h, \boldsymbol{g})}{\dim(\mathcal{G})}$ | Proportion of group attributes $k$ requested by $h$ from each group, averaged over all groups in $\mathcal{G}$ |

**Table 1:** Overview of metrics used to evaluate performance, data use, and consent. We report performance on a held-out test sample. We assume that individuals report group membership to static models, never report group membership to imputed models, and only report to participatory systems when reporting leads to a positive gain. In the latter case, we estimate the gain shown to end-users using a validation set in the training sample.

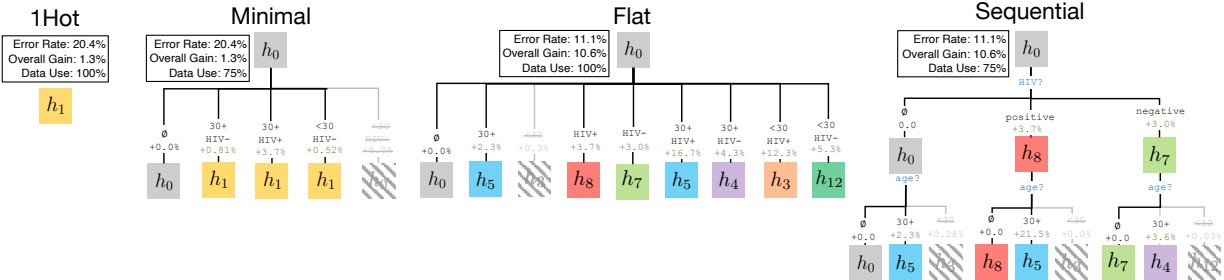

**Figure 4:** Participatory systems for the `saps` dataset. These models predict ICU mortality for groups defined by $\mathcal{G} = \text{HIV} \times \text{age} = [+,-] \times [<30, \ 30+]$. Here, $h_0$ is a generic model, $h_1$ is a 1Hot model fit with a one-hot encoding of $\mathcal{G}$, and $h_2 \cdots h_m$ are 1Hot and mHot models fit for each reporting group. We show the gains of each reporting option above each box, and highlight pruned options in grey. For example, in Seq, group (HIV+, 30+) sees an estimated 21.5% error reduction for `age` after reporting HIV. In contrast, group (HIV+, <30) sees no gain from reporting `age` in addition to HIV status, so this option is pruned.

rather than a logistic regression model (see Appendix D). However, a gain in overall performance does not always translate to gains at the group level. On `saps`, for example, using a random forest also introduces a rationality violation for one group.

**On the Pitfalls of Imputation** One of the simplest approaches to allow individuals to opt out of personalization is to pair a personalized model with an imputation technique. Although this approach can facilitate consent, it does not meet the requirements in 2. Consider a personalized model that exhibits "worsenalization" in Fig. 1. Even if one could correctly impute the group membership for every person, individuals may receive more accurate predictions from a

generic model $h_0$. In practice, imputation is imperfect – as individuals who opt out of reporting their group membership to a personalized model may be assigned "worse" predictions because they are imputed the group membership of a different group. In such cases, opting out may be beneficial, making it difficult for model developers to promote participation while informing consent. Our results highlight the prevalence of these effects across model classes and prediction tasks. For example, on `cardio_eicu` the estimated "risk of imputation" is $-4.6\%$ , indicating that groups can experience an error rate up to $4.6\%$ greater if their values are incorrectly imputed at the level of intersectional groups. The results for KNN-1Hot show that this predicted loss in performance can be realized in practice

| Dataset | Metrics | STATIC | | IMPUTED | | PARTICIPATORY | | |
|---|---|---|---|---|---|---|---|---|
| | | 1Hot | mHot | KNN-1Hot | KNN-mHot | Minimal | Flat | Seq |
| apnea $n = 1152, d = 26$ $\mathcal{G} = \{\texttt{age}, \texttt{sex}\}$ $|\mathcal{G}| = 6$ groups Ustun et al. [55] | Overall Performance | 29.1% | 29.3% | 29.0% | 27.9% | 28.9% | **24.1%** | 24.3% |
| | Overall Gain | 0.1% | -0.1% | 0.2% | 1.3% | 0.3% | **5.1%** | 4.9% |
| | Group Gains | -1.1% – 1.2% | -0.8% – 0.4% | -1.1% – 1.2% | -0.8% – 0.4% | 0.0% – 1.2% | 0.0% – 13.8% | -0.4% – 13.8% |
| | Rat. Violations | **1** | **1** | **1** | **1** | 0 | 0 | 0 |
| | Imputation Risk | -4.9% | -5.2% | | | | | |
| | Options Pruned | 0/6 | 0/6 | 0/12 | 0/12 | 4/7 | 5/12 | 6/12 |
| | Data Use | 100.0% | 100.0% | 0.0% | 0.0% | 33.3% | 83.3% | 58.3% |
| cardio_eicu $n = 1341, d = 49$ $\mathcal{G} = \{\texttt{age}, \texttt{sex}, \texttt{race}\}$ $|\mathcal{G}| = 8$ groups Pollard et al. [43] | Overall Performance | 21.4% | 21.5% | 21.6% | 22.1% | 21.6% | **10.2%** | **10.2%** |
| | Overall Gain | 0.4% | 0.3% | 0.3% | -0.2% | 0.3% | **11.7%** | **11.7%** |
| | Group Gains | -1.3% – 2.6% | -2.7% – 3.0% | -1.3% – 2.6% | -2.7% – 3.0% | 0.0% – 2.6% | 3.1% – 20.9% | 3.1% – 20.9% |
| | Rat. Violations | **1** | **1** | **1** | **1** | 0 | 0 | 0 |
| | Imputation Risk | -4.6% | -5.4% | | | | | |
| | Options Pruned | 0/8 | 0/8 | 0/27 | 0/27 | 6/9 | 10/27 | 9/27 |
| | Data Use | 100.0% | 100.0% | 0.0% | 0.0% | 25.0% | 100.0% | 83.3% |
| cardio_mimic $n = 5289, d = 49$ $\mathcal{G} = \{\texttt{age}, \texttt{sex}, \texttt{race}\}$ $|\mathcal{G}| = 8$ groups Johnson et al. [30] | Overall Performance | 19.4% | 19.3% | 19.3% | 20.1% | 19.2% | **15.7%** | **15.7%** |
| | Overall Gain | -0.1% | -0.0% | -0.0% | -0.8% | 0.1% | **3.5%** | **3.5%** |
| | Group Gains | -0.9% – 0.4% | -0.9% – 0.5% | -0.9% – 0.4% | -0.9% – 0.5% | 0.0% – 0.4% | -1.6% – 9.8% | -1.6% – 9.8% |
| | Rat. Violations | **3** | **2** | **3** | **2** | 0 | **1** | **1** |
| | Imputation Risk | -1.1% | -1.1% | | | | | |
| | Options Pruned | 0/8 | 0/8 | 0/27 | 0/27 | 6/9 | 6/27 | 8/27 |
| | Data Use | 100.0% | 100.0% | 0.0% | 0.0% | 25.0% | 100.0% | 91.7% |
| coloncancer $n = 29211, d = 72$ $\mathcal{G} = \{\texttt{age}, \texttt{sex}\}$ $|\mathcal{G}| = 6$ groups Scosyrev et al. [45] | Overall Performance | 37.0% | 36.7% | 37.0% | 36.9% | 37.0% | 36.6% | **36.1%** |
| | Overall Gain | 0.1% | 0.4% | 0.1% | 0.2% | 0.1% | 0.5% | **1.0%** |
| | Group Gains | -0.4% – 0.3% | -0.1% – 1.1% | -0.4% – 0.3% | -0.1% – 1.1% | 0.0% – 0.3% | 0.0% – 1.7% | 0.2% – 1.7% |
| | Rat. Violations | **1** | 0 | **1** | 0 | 0 | 0 | 0 |
| | Imputation Risk | -1.4% | -0.9% | | | | | |
| | Options Pruned | 0/6 | 0/6 | 0/12 | 0/12 | 5/7 | 7/12 | 5/12 |
| | Data Use | 100.0% | 100.0% | 0.0% | 0.0% | 16.7% | 50.0% | 75.0% |
| lungcancer $n = 120641, d = 84$ $\mathcal{G} = \{\texttt{age}, \texttt{sex}\}$ $|\mathcal{G}| = 6$ groups Scosyrev et al. [45] | Overall Performance | 19.6% | 19.6% | 19.9% | 19.8% | 19.5% | 18.9% | **18.9%** |
| | Overall Gain | -0.1% | -0.1% | -0.3% | -0.2% | 0.0% | 0.6% | **0.6%** |
| | Group Gains | -0.4% – 0.2% | -0.3% – 0.2% | -0.4% – 0.2% | -0.3% – 0.2% | 0.0% – 0.0% | 0.0% – 0.9% | 0.3% – 0.9% |
| | Rat. Violations | **4** | **4** | **4** | **4** | 0 | 0 | 0 |
| | Imputation Risk | -0.5% | -0.5% | | | | | |
| | Options Pruned | 0/6 | 0/6 | 0/12 | 0/12 | 6/7 | 3/12 | 7/12 |
| | Data Use | 100.0% | 100.0% | 0.0% | 0.0% | 0.0% | 83.3% | 58.3% |
| saps $n = 7797, d = 36$ $\mathcal{G} = \{\texttt{HIV}, \texttt{age}\}$ $|\mathcal{G}| = 4$ groups Allyn et al. [3] | Overall Performance | 20.4% | 20.7% | 20.4% | 29.4% | 20.4% | **11.1%** | 11.1% |
| | Overall Gain | 1.3% | 1.0% | 1.3% | -7.7% | 1.3% | **10.6%** | 10.6% |
| | Group Gains | 0.0% – 3.6% | 0.0% – 2.7% | 0.0% – 3.6% | 0.0% – 2.7% | 0.0% – 3.6% | 4.3% – 17.2% | 4.3% – 17.2% |
| | Rat. Violations | 0 | 0 | 0 | 0 | 0 | 0 | 0 |
| | Imputation Risk | 0.0% | -2.4% | | | | | |
| | Options Pruned | 0/4 | 0/4 | 0/9 | 0/9 | 1/5 | 1/9 | 3/9 |
| | Data Use | 100.0% | 100.0% | 0.0% | 0.0% | 75.0% | 100.0% | 75.0% |

**Table 2:** Participatory systems and personalized models for all datasets. We summarize metrics in Table 1 and present results for other model classes and prediction tasks in Appendix D.

using KNN-imputation, as we find that the imputed system leads to rationality violations on 5/6 datasets.

## 5. Discussion and Limitations

In this work, we introduced a family of prediction models that allow individuals to report personal data at prediction time. These systems can inform consent while producing large improvements in performance and data use for each group that reports personal data.

One common concern is that allowing individuals to opt out of personalization may preclude developers from collecting data to improve performance in later rounds of retraining. In this case, the underlying principle for data collection is *purpose specification* [39]. If the purpose of data collection is to monitor or improve a model, then individuals could be

asked to report information voluntarily for this purpose. If data collection aims to improve the model, then individuals are within their right to opt out.

In practice, the viability of reaping the benefits of personalization hinges on the individual preferences for disclosure, which can change based on the information solicited, the outcome predicted, and the ability to inform users effectively of these impacts [5]. Implementing these systems will require developing tailored approaches to communicate the gains of personalization (e.g., communicating risk and uncertainty).

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

# Participatory Personalization in Classification

## Supplementary Material

# A. Supporting Material for Section 2 – Participatory Systems

## A.1. Agent Model for Individual Disclosure

The performance of participatory systems will depend on individual reporting decisions. In what follows, we characterize how participatory systems will perform under a generalized model of individual disclosure. Given a participatory system $h : \mathcal{X} \times \mathcal{R} \to \mathcal{Y}$, we assume that each person will report group membership as:

$$\boldsymbol{r}_i \in \underset{\boldsymbol{r} \in \mathcal{R}}{\operatorname{argmax}}\, u_i(\boldsymbol{r}; h)$$

Here, the utility function can be

$$u_i(\boldsymbol{r}; h) = b_i(\boldsymbol{r}; h) - c_i(\boldsymbol{r}),$$

where $c_i(\cdot)$ and $b_i(\cdot)$ denote their cost and benefit of disclosure, respectively. We assume that costs increase monotonically with information that is disclosed so that $c_i(\boldsymbol{r}) \geq 0$ for all $\boldsymbol{r} \in \mathcal{R}$ and $c_i(\boldsymbol{r}) \leq c_i(\boldsymbol{r}')$ for $\boldsymbol{r} \subseteq \boldsymbol{r}'$. We assume that benefits increase monotonically with true risk so that $b_i(\boldsymbol{r}, h) > b_i(\boldsymbol{r}', h)$ when $R_{\boldsymbol{r}}(h(\boldsymbol{x}_i, \boldsymbol{r})) < R_{\boldsymbol{r}}(h(\boldsymbol{x}_i, \boldsymbol{r}'))$.

The following remarks apply to any participatory system $f : \mathcal{X} \times \mathcal{R} \to Y$ that include a personalized model $h : \mathcal{X} \times \mathcal{G} \to \mathcal{Y}$ and a generic model $h_0 : \mathcal{X} \to \mathcal{Y}$ as its components.

- Every participatory system $f$ will perform as well as a generic model $h_0$. When a personalized model $h$ requires users to report information detrimental to performance (see Fig. 1), individuals incur a cost of disclosure without receiving a benefit. In such instances, a minimal system $f : \mathcal{X} \times \mathcal{R}^{\min} \to Y$ would allow individuals to opt out of detrimental personalization and receive predictions from a generic model.

- Every participatory system $f$ with more reporting options will perform better. Given that utility can only increase with the number of reporting options, the maximum utility for each person will exceed that of a minimal system. Thus, flat and sequential systems will perform better than a minimal system.

- The best-case performance of any participatory system will exceed the performance of any of its components. Thus, we are guaranteed that any participatory system will outperform a traditional personalized model so long as it is considered a component.

## A.2. Profiling System Performance with Respect to Participation

We can use the models for individual disclosure to evaluate how a participatory system will perform once it is deployed. Given a participatory system, we can conduct this evaluation by simulating the parameters in the individual disclosure model shown above. We can then summarize the results from this evaluation for each intersectional group through a performance profile that shows how the system performance will vary across different levels of participation.

We show performance profiles for participatory systems built for the `saps` dataset in Fig. 3. Here, we measure the benefit of disclosure in terms of their expected performance gain and simulate the cost of reporting for each individual by sampling their reporting cost from a uniform distribution – i.e., for each individual $i$, we sample $c_i$ as $c_i \sim \text{Uniform}(0, \gamma)$, where $\gamma \in [0, 0.2]$. For each value of $\gamma$, we sample reporting costs 10 times and average over the per group performance error for each sampled cost.

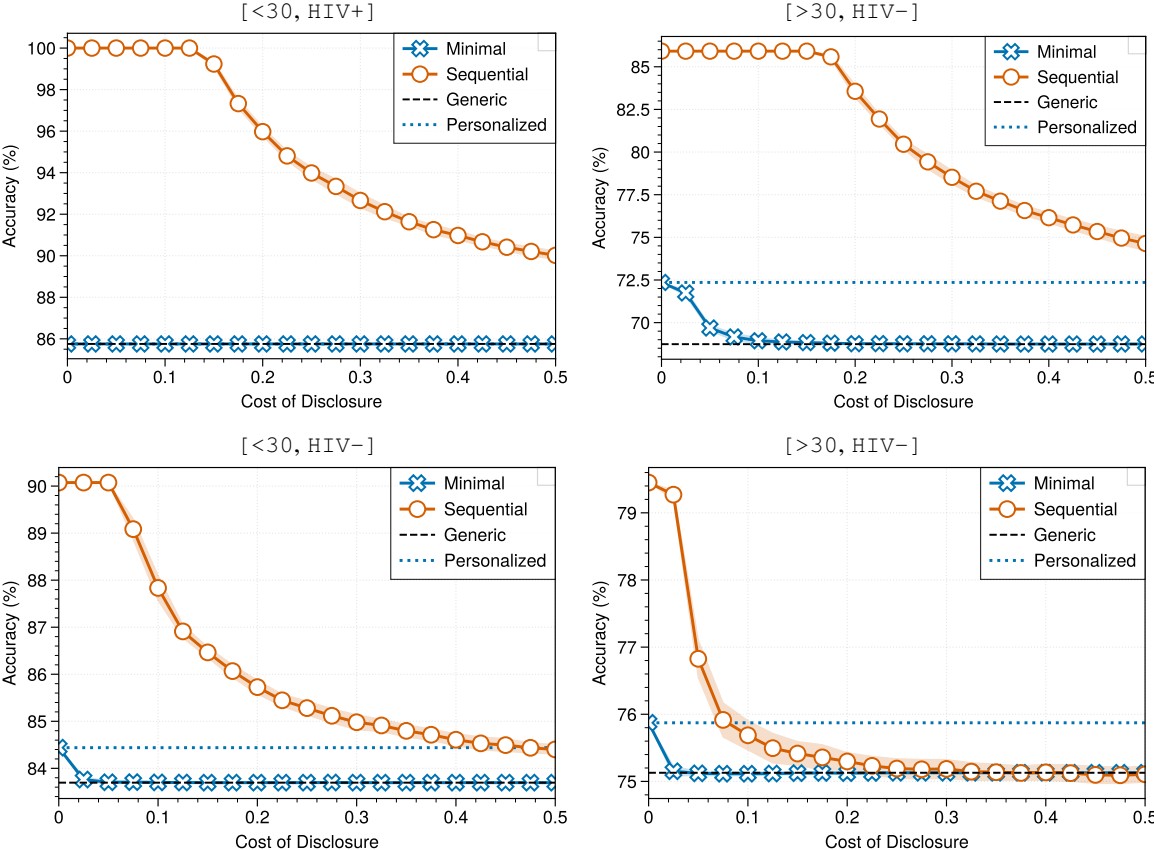

**Figure 5:** Performance profiles of the simulations performed for each intersectional group in the `saps` dataset. The sequential system outperforms static personalized systems when all group attributes are reported. When the cost of reporting is high, the sequential system still outperforms minimally personalized systems as evidenced by higher accuracy at varying reporting cost thresholds.

# B. Supporting Material for Section 3 – Learning Participatory Systems

## B.1. Enumeration Routine for Algorithm 1

We summarize the Enumeration routine in Algorithm 2. Algorithm 2 takes as input a set of group attributes $\mathcal{G}$ and a dataset $\mathcal{D}$ and outputs a collection of reporting interfaces $\mathbb{T}$ that obey ordering and plausibility constraints.

---

**Algorithm 2** Enumerate All Possible Reporting Trees for Reporting Options $\mathcal{G}$

---

 1: **procedure** VIABLETREES($\mathcal{G}, \mathcal{D}$)
 2:      **if** $\dim(\mathcal{G}) = 1$ **return** $[T_{\mathcal{G}}]$                             *base case: we are left with only a single attribute on which to branch*
 3:      $\mathbb{T} \leftarrow [\,]$
 4:      **for** each group attribute $\mathcal{A} \in [\mathcal{G}_1, \ldots, \mathcal{G}_k]$ **do**
 5:          $T_{\mathcal{A}} \leftarrow$ reporting tree of depth 1 with $|\mathcal{A}|$ leaves
 6:          $\mathcal{S} \leftarrow$ ViableTrees($\mathcal{G} \setminus \mathcal{A}, \mathcal{D}$)                           *all subtrees using all attributes except $\mathcal{A}$*
 7:          **for** $\Pi$ in ValidAssignments($\mathcal{S}, \mathcal{A}, \mathcal{D}$) **do**:       *each assignment is a permutation of $|\mathcal{A}|$ to leaves of $T_{\mathcal{A}}$*
 8:              $\mathbb{T} \leftarrow \mathbb{T} \cup T_{\mathcal{A}}.\mathsf{assign}(\Pi)$                   *extends the tree by assigning subtrees to each leaf*
 9:          **end for**
10:      **end for**
11:      **return** $\mathbb{T}$, reporting interfaces for group attributes $\mathcal{G}$ that obey plausibility and ordering constraints
12: **end procedure**

---

The routine enumerates all possible reporting interfaces for a given set of group attributes $\mathcal{G}$ through a recursive branching process. Given a set of group attributes, the routine is called for each attribute that has yet to be considered in the tree Line 4, ensuring a complete enumeration. We note that the routine is only called for building Sequential systems since there is only one possible reporting interface for Minimal and Flat systems.

Enumerating all possible trees ensures we can recover the best tree given the selection criteria and allows practitioners to choose between models based on other criteria. We generate trees that meet plausibility constraints based on the dataset, such as having at least one negative and one positive sample and at least $s$ total samples at each leaf. In settings constrained by computational resources, we can impose additional stopping criteria and modify the ordering to enumerate more plausible trees first or exclusively (e.g., by changing the ordering of $\mathcal{G}$ or imposing constraints in VALIDASSIGNMENTS).

## B.2. Assignment Routine for Algorithm 1

We summarize the routine for AssignModels procedure in Algorithm 3.

---

**Algorithm 3** Assigning Models

---

 1: **procedure** ASSIGNMODELS($T, \mathcal{M}, \mathcal{D}$)
 2:      $Q \leftarrow [T.\mathsf{root}]$                                             *initialize with the root of the tree, reporting group $\varnothing$*
 3:      **while** $Q$ is not empty **do**
 4:          $\boldsymbol{r} \leftarrow Q.\mathsf{pop}()$
 5:          $\mathcal{M}_{\boldsymbol{r}} \leftarrow$ ViableModels($\mathcal{M}, \boldsymbol{r}$)                         *filter $\mathcal{M}$ to models that can be assigned to $\boldsymbol{r}$*
 6:          $h^* \leftarrow \underset{h \in \mathcal{M}_{\boldsymbol{r}}}{\arg\min}\, \hat{R}_{\boldsymbol{r}}(h, \mathcal{D})$              *assign the model with the best training performance*
 7:          $T.\mathsf{set\_model}(\boldsymbol{r}, h^*)$
 8:          **for** $\boldsymbol{r}' \in T.\mathsf{get\_subgroups}(\boldsymbol{r})$ **do**             *iterate through the children reporting groups of $\boldsymbol{r}$*
 9:              $Q.\mathsf{enqueue}(\boldsymbol{r}')$
10:          **end for**
11:      **end while**
12:      **return** $T$ that maximizes gain for each reporting group
13: **end procedure**

---

Algorithm 3 takes as inputs a reporting tree $T$, a pool candidate models $\mathcal{M}$, and an assignment (training) dataset $\mathcal{D}$ and outputs a tree $T$ that maximizes the gains of reporting group information. The pool of candidate models is filtered to viable models for each reporting group. Since the pool of candidate models includes the generic model $h_0$, each reporting group will have at least one viable model. We assign each reporting group the best-performing model on the training set and default to the generic model $h_0$ when a better-performing personalized model is not found. We assign performance on the training set and then prune using performance on the validation set to avoid biased gain estimations.

## B.3. Pruning Routine for Algorithm 1

We summarize the routine used for the PruneLeaves procedure in Algorithm 1. The PruneLeaves routine

---

**Algorithm 4** Pruning Participatory Systems

---

1: **procedure** PRUNELEAVES($T, \mathcal{D}$)
2:  $Stack \leftarrow [T.\text{leaves}]$                          *initialize stack with all leaves*
3:  **repeat**
4:    $\boldsymbol{r} \leftarrow Stack.pop()$
5:    $h \leftarrow T.\text{get\_model}(\boldsymbol{r})$
6:    $h' \leftarrow T.\text{get\_model}(\text{pa}(\boldsymbol{r}))$
7:    **if** not $\text{Test}(\boldsymbol{r}, h, h', \mathcal{D})$ **then**          *test gains to see if parent model is as good as leaf model*
8:      $T.\text{prune}(\boldsymbol{r})$
9:    **end if**
10:   **if** $T.\text{get\_children}(\text{pa}(\boldsymbol{r}))$ is empty **then**    *consider pruning the parent if the parent has become a leaf*
11:     $Stack.\text{enqueue}(\text{pa}(\boldsymbol{r}))$
12:   **end if**
13:  **until** $Stack$ is empty
14:  **return** $T$, reporting interface that ensures data collection leads to gain
15: **end procedure**

---

Algorithm 1 takes as input a reporting interface $T$ and a validation sample $\mathcal{D}$, and performs a bottom-up pruning to output a reporting interface $T$ that asks individuals to report attributes that are expected to lead to a gain. The pruning decision at each leaf is based on a hypothesis test that evaluates the gains of reporting for a reporting group on a validation dataset. This test has the form:

$$H_0 : R_{\boldsymbol{g}}(h) \leq R_{\boldsymbol{g}}(h') \quad \text{vs.} \quad H_A : R_{\boldsymbol{g}}(h) > R_{\boldsymbol{g}}(h')$$

This procedure evaluates the gains of reporting by comparing the performance of a model assigned at a leaf node $h$ and a model assigned at a parent node $h'$ which does not use the reported information. Here, the null hypothesis $H_0$ assumes that the parent model performs as well as the leaf model – and thus, we reject the null hypothesis when there is sufficient evidence to suggest that reporting will improve performance in deployment. Our routine allows practitioners to specify the hypothesis test to compute the gains. By default, we use the McNemar test for accuracy [21] and the Delong test for AUC [19, 50]. In general, we can use a bootstrap hypothesis test [20].

## B.4. Greedy Induction of Sequential Reporting Interface

We present an additional routine to construct reporting interfaces for sequential systems in Algorithm 5. We include this routine as an alternative option that can be used to construct a reporting interface in settings where it may be impractical or undesirable to enumerate all possible reporting interfaces. The procedure results in a valid reporting interface that ensures gains. However, it does not guarantee an optimal tree in terms of maximizing the overall gain and does not allow to practitioners to choose between reporting interfaces after training.

---

**Algorithm 5** Greedy Induction Routine for Sequential Reporting Interfaces

---

1: **procedure** GREEDYTREE($\mathcal{R}$)
2:  $T \leftarrow$ empty reporting interface
3:  **repeat**
4:    **for** $\boldsymbol{r} \in \text{leaves}(T)$ **do**
5:     $\{\mathcal{A}_{\boldsymbol{r}}\} \leftarrow G_i : \boldsymbol{r}[i] = \varnothing$          *$\{\mathcal{A}_{\boldsymbol{r}}\}$ contains all heretofore unused attributes*
6:     $\mathcal{A}^* \leftarrow \text{argmax}_{\mathcal{A} \in \{\mathcal{A}_{\boldsymbol{r}}\}} \min_{\boldsymbol{r}' \in \boldsymbol{r}.\text{split}(\mathcal{A})} \Delta_{\boldsymbol{r}'}(\boldsymbol{r}', \boldsymbol{r})$
7:     $\boldsymbol{r}.\text{split}(\mathcal{A}^*)$                  *Split on attribute that maximizes worse-case gain*
8:    **end for**
9:  **until** no splits are added
10:  **return** $T$, reporting interface that ensures gains for reporting each $\mathcal{R}$.
11: **end procedure**

---

Algorithm 5 takes as input a collection of reporting options $\mathcal{R}$ and outputs a single reporting interface using a greedy tree induction routine that chooses the attribute to report to maximize the minimum gain at each step. The procedure uses the reporting options to iteratively construct a reporting tree that branches on all of the attributes in $\mathcal{R}$. The procedure considers each unused attribute for each splitting point and splits on the attribute that provides the greatest minimum gain for the groups contained at that node.

## C. Description of Datasets used in Section 4 – Experiments

We include additional information about the datasets used in Section 4.

| Dataset | Reference | Outcome Variable | $n$ | $d$ | $m$ | $\mathcal{G}$ |
|---------|-----------|------------------|-----|-----|-----|---------------|
| apnea | Ustun et al. [55] | patient has obstructive sleep apnea | 1,152 | 28 | 6 | {age, sex} |
| cardio_eicu | Pollard et al. [43] | patient with cardiogenic shock dies | 1,341 | 49 | 8 | {age, sex, race} |
| cardio_mimic | Johnson et al. [30] | patient with cardiogenic shock dies | 5,289 | 49 | 8 | {age, sex, race} |
| coloncancer | Scosyrev et al. [45] | patient dies within 5 years | 29,211 | 72 | 6 | {age, sex} |
| lungcancer | Scosyrev et al. [45] | patient dies within 5 years | 120,641 | 84 | 6 | {age, sex} |
| saps | Allyn et al. [3] | ICU mortality | 7,797 | 36 | 4 | {age, HIV} |

**Table 3:** Overview of datasets used to fit clinical prediction models in Section 4. Here: $n$ denotes the number of examples in each dataset; $d$ denotes the number of features; $\mathcal{G}$ denotes the group attributes that are used for personalization; and $m = |\mathcal{G}|$ denotes the number of intersectional groups. Each dataset is de-identified and available to the public. The cardio_eicu, cardio_mimic, lungcancer datasets require access to public repositories listed under the references. The saps and apnea datasets must be requested from the authors. The support dataset can be downloaded directly from the URL below.

**apnea**   We use the obstructive sleep apnea (OSA) dataset outlined in Ustun et al. [55]. This dataset includes a cohort of 1,152 patients where 23% have OSA. We use all available features (e.g. BMI, comorbidities, age, and sex) and binarize them, resulting in 26 binary features.

**cardio_eicu & cardio_mimic**   Cardiogenic shock is an acute condition in which the heart cannot provide sufficient blood to the vital organs [29]. These datasets are designed to predict cardiogenic shock for patients in intensive care. Each dataset contains the same features, group attributes, and outcome variables for patients in different cohorts. The cardio_eicu dataset contains records for a cohort of patients in the Collaborative Research Database V2.0 [43]. The cardio_eicu dataset contains records for a cohort of patients in the MIMIC-III [30] database. Here, the outcome variable indicates whether a patient in the ICU with cardiogenic shock will die while in the ICU. The features encode the results of vital signs and routine lab tests (e.g. systolic BP, heart rate, hemoglobin count) that were collected up to 24 hours before the onset of cardiogenic shock.

**lungcancer**   We consider a cohort of 120,641 patients who were diagnosed with lung cancer between 2004-2016 and monitored as part of the National Cancer Institute SEER study [45]. Here, the outcome variable indicates if a patient dies within five years from any cause, and 16.9% of patients died within the first five years from diagnosis. The cohort includes patients from Greater California, Georgia, Kentucky, New Jersey, and Louisiana, and does not cover patients who were lost to follow-up (censored). Age and Sex were considered as group attributes. The features reflect the morphology and histology of the tumor (e.g., size, metastasis, stage, node count and location, number and location of notes) as well as interventions that were administered at the time of diagnosis (e.g., surgery, chemo, radiology).

**coloncancer**   We consider a cohort of 120,641 patients who were diagnosed with colorectal cancer between 2004-2016 and monitored as part of the National Cancer Institute SEER study [45]. Here, the outcome variable indicates if a patient dies within five years from any cause, and 42.1% of patients die within the first five years from diagnosis. The cohort includes patients from Greater California. Age and Sex were considered as group attributes. The features reflect the morphology and histology of the tumor (e.g., size, metastasis, stage, node count and location, number and location of notes) as well as interventions that were administered at the time of diagnosis (e.g., surgery, chemo, radiology).

**saps**   The Simplified Acute Physiology Score II (SAPS II) score predicts the risk of mortality of critically-ill patients in intensive care [35]. The data contains records of 7,797 patients from 137 medical centers in 12 countries. Here, the outcome variable indicates whether a patient dies in the ICU, with 12.8% patient of patients dying. The features reflect comorbidities, vital signs, and lab measurements.

# D. Experimental Results for Model Classes and Prediction Tasks

In this Appendix, we present experimental results for additional model classes and prediction tasks. We produce these results using the setup in Section 4.1, and summarize them in the same way as Table 2. We refer to them in our discussion in Section 4.2.

## D.1. Logistic Regression for Ranking (AUC)

| Dataset | Metrics | STATIC | | IMPUTED | | PARTICIPATORY | | |
|---|---|---|---|---|---|---|---|---|
| | | 1Hot | mHot | KNN-1Hot | KNN-mHot | Minimal | Flat | Seq |
| apnea $n = 1152, d = 26$ $\mathcal{G} = \{\texttt{age}, \texttt{sex}\}$ $|\mathcal{G}| = 6$ groups Ustun et al. [55] | Overall Performance | 0.774 | 0.774 | 0.776 | 0.776 | 0.776 | **0.851** | **0.851** |
| | Overall Gain | -0.002 | -0.002 | 0.000 | -0.000 | 0.000 | **0.074** | **0.074** |
| | Group Gains | -0.002 – 0.002 | -0.002 – 0.003 | -0.002 – 0.002 | -0.002 – 0.003 | 0.000 – 0.002 | 0.004 – 0.115 | 0.004 – 0.115 |
| | Max Disparity | 0.004 | 0.005 | 0.004 | 0.005 | 0.002 | 0.111 | 0.111 |
| | Rat. Violations | **2** | **2** | **2** | **2** | 0 | 0 | 0 |
| | Imputation Risk | -0.002 | -0.002 | | | | | |
| | Options Pruned | 0/6 | 0/6 | 0/12 | 0/12 | 5/7 | 4/12 | 4/12 |
| | Data Use | 100.0% | 100.0% | 0.0% | 0.0% | 16.7% | 100.0% | 83.3% |
| cardio_eicu $n = 1341, d = 49$ $\mathcal{G} = \{\texttt{age}, \texttt{sex}, \texttt{race}\}$ $|\mathcal{G}| = 8$ groups Pollard et al. [43] | Overall Performance | 0.864 | 0.863 | 0.863 | 0.862 | 0.865 | 0.966 | **0.966** |
| | Overall Gain | 0.002 | 0.001 | 0.000 | -0.001 | 0.002 | 0.103 | **0.103** |
| | Group Gains | -0.005 – 0.003 | -0.010 – 0.010 | -0.005 – 0.003 | -0.010 – 0.010 | 0.000 – 0.003 | 0.010 – 0.180 | 0.010 – 0.180 |
| | Max Disparity | 0.009 | 0.019 | 0.009 | 0.019 | 0.003 | 0.170 | 0.170 |
| | Rat. Violations | **3** | **3** | **3** | **3** | 0 | 0 | 0 |
| | Imputation Risk | -0.005 | -0.010 | | | | | |
| | Options Pruned | 0/8 | 0/8 | 0/27 | 0/27 | 6/9 | 13/27 | 11/27 |
| | Data Use | 100.0% | 100.0% | 0.0% | 0.0% | 25.0% | 100.0% | 95.8% |
| cardio_mimic $n = 5289, d = 49$ $\mathcal{G} = \{\texttt{age}, \texttt{sex}, \texttt{race}\}$ $|\mathcal{G}| = 8$ groups Johnson et al. [30] | Overall Performance | 0.881 | 0.881 | 0.882 | 0.880 | 0.881 | **0.914** | **0.914** |
| | Overall Gain | 0.000 | 0.000 | 0.002 | -0.000 | 0.000 | **0.034** | **0.034** |
| | Group Gains | -0.001 – 0.001 | -0.001 – 0.001 | -0.001 – 0.001 | -0.001 – 0.001 | 0.000 – 0.001 | 0.008 – 0.057 | 0.008 – 0.057 |
| | Max Disparity | 0.002 | 0.002 | 0.002 | 0.002 | 0.001 | 0.049 | 0.049 |
| | Rat. Violations | **3** | **3** | **3** | **3** | 0 | 0 | 0 |
| | Imputation Risk | -0.001 | -0.001 | | | | | |
| | Options Pruned | 0/8 | 0/8 | 0/27 | 0/27 | 6/9 | 9/27 | 8/27 |
| | Data Use | 100.0% | 100.0% | 0.0% | 0.0% | 25.0% | 100.0% | 91.7% |
| coloncancer $n = 29211, d = 72$ $\mathcal{G} = \{\texttt{age}, \texttt{sex}\}$ $|\mathcal{G}| = 6$ groups Scosyrev et al. [45] | Overall Performance | 0.685 | 0.685 | 0.683 | 0.683 | 0.685 | **0.700** | 0.700 |
| | Overall Gain | 0.001 | 0.002 | -0.000 | -0.000 | 0.001 | **0.016** | 0.016 |
| | Group Gains | -0.001 – 0.002 | -0.001 – 0.001 | -0.001 – 0.002 | -0.001 – 0.001 | 0.000 – 0.001 | 0.001 – 0.021 | 0.001 – 0.021 |
| | Max Disparity | 0.003 | 0.002 | 0.003 | 0.002 | 0.001 | 0.020 | 0.020 |
| | Rat. Violations | **3** | **2** | **3** | **2** | 0 | 0 | 0 |
| | Imputation Risk | -0.001 | -0.002 | | | | | |
| | Options Pruned | 0/6 | 0/6 | 0/12 | 0/12 | 5/7 | 2/12 | 5/12 |
| | Data Use | 100.0% | 100.0% | 0.0% | 0.0% | 16.7% | 100.0% | 75.0% |
| lungcancer $n = 120641, d = 84$ $\mathcal{G} = \{\texttt{age}, \texttt{sex}\}$ $|\mathcal{G}| = 6$ groups Scosyrev et al. [45] | Overall Performance | 0.855 | 0.855 | 0.852 | 0.854 | 0.855 | **0.861** | 0.861 |
| | Overall Gain | 0.001 | 0.001 | -0.002 | 0.000 | 0.001 | **0.006** | 0.006 |
| | Group Gains | -0.000 – 0.000 | -0.000 – 0.000 | -0.000 – 0.000 | -0.000 – 0.000 | 0.000 – 0.000 | 0.001 – 0.012 | 0.001 – 0.012 |
| | Max Disparity | 0.001 | 0.001 | 0.001 | 0.001 | 0.000 | 0.011 | 0.011 |
| | Rat. Violations | **2** | **2** | **2** | **2** | 1 | 0 | 0 |
| | Imputation Risk | -0.000 | -0.000 | | | | | |
| | Options Pruned | 0/6 | 0/6 | 0/12 | 0/12 | 4/7 | 2/12 | 2/12 |
| | Data Use | 100.0% | 100.0% | 0.0% | 0.0% | 33.3% | 100.0% | 91.7% |
| saps $n = 7797, d = 36$ $\mathcal{G} = \{\texttt{HIV}, \texttt{age}\}$ $|\mathcal{G}| = 4$ groups Allyn et al. [3] | Overall Performance | 0.875 | 0.877 | 0.875 | 0.857 | 0.875 | **0.960** | 0.960 |
| | Overall Gain | 0.010 | 0.011 | 0.010 | -0.008 | 0.009 | **0.095** | 0.095 |
| | Group Gains | -0.000 – 0.016 | -0.002 – 0.019 | -0.000 – 0.016 | -0.002 – 0.019 | 0.000 – 0.016 | 0.035 – 0.141 | 0.035 – 0.141 |
| | Max Disparity | 0.017 | 0.021 | 0.017 | 0.021 | 0.016 | 0.106 | 0.106 |
| | Rat. Violations | **1** | **1** | **1** | **1** | 0 | 0 | 0 |
| | Imputation Risk | -0.000 | -0.002 | | | | | |
| | Options Pruned | 0/4 | 0/4 | 0/9 | 0/9 | 1/5 | 2/9 | 3/9 |
| | Data Use | 100.0% | 100.0% | 0.0% | 0.0% | 75.0% | 100.0% | 87.5% |

**Table 4:** Overview of performance, data use, and consent for all personalized models and systems on all datasets as measured by **test auc**. We show the performance of models and systems built using **logistic regression**.

## D.2. Random Forests for Decision-Making (Error)

| Dataset | Metrics | STATIC | | IMPUTED | | PARTICIPATORY | | |
|---|---|---|---|---|---|---|---|---|
| | | 1Hot | mHot | KNN-1Hot | KNN-mHot | Minimal | Flat | Seq |
| `apnea` $n = 1152, d = 26$ $\mathcal{G} = \{\texttt{age}, \texttt{sex}\}$ $|\mathcal{G}| = 6$ groups Ustun et al. [55] | Overall Performance | 26.3% | 26.0% | 25.9% | 27.4% | 26.3% | **12.2%** | **12.2%** |
| | Overall Gain | 1.5% | 1.8% | 1.9% | 0.4% | 1.5% | **15.6%** | **15.6%** |
| | Group Gains | -0.8% − 4.2% | 0.4% − 3.8% | -0.8% − 4.2% | 0.4% − 3.8% | 0.0% − 4.2% | 5.3% − 22.2% | 5.3% − 22.2% |
| | Max Disparity | 5.0% | 3.4% | 5.0% | 3.4% | 4.2% | 16.9% | 16.9% |
| | Rat. Violations | **1** | 0 | **1** | 0 | 0 | 0 | 0 |
| | Imputation Risk | -1.2% | -1.2% | | | | | |
| | Options Pruned | 0/6 | 0/6 | 0/12 | 0/12 | 2/7 | 1/12 | 2/12 |
| | Data Use | 100.0% | 100.0% | 0.0% | 0.0% | 66.7% | 100.0% | 91.7% |
| `cardio_eicu` $n = 1341, d = 49$ $\mathcal{G} = \{\texttt{age}, \texttt{sex}, \texttt{race}\}$ $|\mathcal{G}| = 8$ groups Pollard et al. [43] | Overall Performance | 18.6% | 17.8% | 18.2% | 18.6% | 18.4% | **5.7%** | 6.0% |
| | Overall Gain | -0.2% | 0.6% | 0.2% | -0.2% | 0.0% | **12.7%** | 12.4% |
| | Group Gains | -3.5% − 1.4% | -2.2% − 3.0% | -3.5% − 1.4% | -2.2% − 3.0% | 0.0% − 0.0% | 6.0% − 14.9% | 6.0% − 14.9% |
| | Max Disparity | 4.9% | 5.3% | 4.9% | 5.3% | 0.0% | 8.9% | 8.9% |
| | Rat. Violations | **2** | **2** | **2** | **2** | 0 | 0 | 0 |
| | Imputation Risk | -3.5% | -2.2% | | | | | |
| | Options Pruned | 0/8 | 0/8 | 0/27 | 0/27 | 8/9 | 11/27 | 8/27 |
| | Data Use | 100.0% | 100.0% | 0.0% | 0.0% | 0.0% | 100.0% | 91.7% |
| `cardio_mimic` $n = 5289, d = 49$ $\mathcal{G} = \{\texttt{age}, \texttt{sex}, \texttt{race}\}$ $|\mathcal{G}| = 8$ groups Johnson et al. [30] | Overall Performance | 19.9% | 20.1% | 19.9% | 20.2% | 19.6% | 11.5% | **11.4%** |
| | Overall Gain | -0.3% | -0.5% | -0.3% | -0.6% | 0.0% | 8.1% | **8.1%** |
| | Group Gains | -1.1% − 1.3% | -1.3% − 0.5% | -1.1% − 1.3% | -1.3% − 0.5% | 0.0% − 0.0% | 1.0% − 14.9% | 1.0% − 14.9% |
| | Max Disparity | 2.4% | 1.7% | 2.4% | 1.7% | 0.0% | 13.8% | 13.8% |
| | Rat. Violations | **5** | **6** | **5** | **6** | 0 | 0 | 0 |
| | Imputation Risk | -1.1% | -1.3% | | | | | |
| | Options Pruned | 0/8 | 0/8 | 0/27 | 0/27 | 8/9 | 6/27 | 5/27 |
| | Data Use | 100.0% | 100.0% | 0.0% | 0.0% | 0.0% | 100.0% | 87.5% |
| `coloncancer` $n = 29211, d = 72$ $\mathcal{G} = \{\texttt{age}, \texttt{sex}\}$ $|\mathcal{G}| = 6$ groups Scosyrev et al. [45] | Overall Performance | 37.2% | 37.0% | 37.2% | 37.0% | 37.0% | **35.9%** | 35.9% |
| | Overall Gain | -0.2% | 0.0% | -0.2% | -0.0% | 0.0% | **1.0%** | 1.0% |
| | Group Gains | -0.7% − 0.1% | -0.3% − 0.2% | -0.7% − 0.1% | -0.3% − 0.2% | 0.0% − 0.0% | 0.1% − 3.2% | 0.1% − 3.2% |
| | Max Disparity | 0.7% | 0.5% | 0.7% | 0.5% | 0.0% | 3.1% | 3.1% |
| | Rat. Violations | **4** | **1** | **4** | **1** | 0 | 0 | 0 |
| | Imputation Risk | -0.7% | -0.3% | | | | | |
| | Options Pruned | 0/6 | 0/6 | 0/12 | 0/12 | 6/7 | 3/12 | 5/12 |
| | Data Use | 100.0% | 100.0% | 0.0% | 0.0% | 0.0% | 100.0% | 75.0% |
| `lungcancer` $n = 120641, d = 84$ $\mathcal{G} = \{\texttt{age}, \texttt{sex}\}$ $|\mathcal{G}| = 6$ groups Scosyrev et al. [45] | Overall Performance | 20.0% | 20.2% | 20.0% | 20.3% | 20.0% | **19.3%** | 19.3% |
| | Overall Gain | 0.1% | -0.1% | 0.1% | -0.2% | 0.1% | **0.8%** | 0.7% |
| | Group Gains | -0.3% − 0.2% | -0.5% − 0.0% | -0.3% − 0.2% | -0.5% − 0.0% | 0.0% − 0.2% | 0.0% − 2.3% | 0.0% − 2.2% |
| | Max Disparity | 0.6% | 0.5% | 0.6% | 0.5% | 0.2% | 2.3% | 2.1% |
| | Rat. Violations | **1** | **4** | **1** | **4** | 0 | 0 | 0 |
| | Imputation Risk | -0.3% | -0.5% | | | | | |
| | Options Pruned | 0/6 | 0/6 | 0/12 | 0/12 | 3/7 | 1/12 | 3/12 |
| | Data Use | 100.0% | 100.0% | 0.0% | 0.0% | 50.0% | 100.0% | 83.3% |
| `saps` $n = 7797, d = 36$ $\mathcal{G} = \{\texttt{HIV}, \texttt{age}\}$ $|\mathcal{G}| = 4$ groups Allyn et al. [3] | Overall Performance | 14.1% | 15.0% | 14.1% | 15.7% | 13.9% | **9.8%** | **9.8%** |
| | Overall Gain | 0.9% | -0.0% | 0.9% | -0.7% | 1.1% | **5.2%** | **5.2%** |
| | Group Gains | -0.8% − 3.4% | -0.5% − 0.3% | -0.8% − 3.4% | -0.5% − 0.3% | 0.0% − 3.4% | 0.0% − 16.4% | 0.0% − 16.4% |
| | Max Disparity | 4.2% | 0.8% | 4.2% | 0.8% | 3.4% | 16.4% | 16.4% |
| | Rat. Violations | **1** | **1** | **1** | **1** | 0 | 0 | 0 |
| | Imputation Risk | -0.8% | -0.7% | | | | | |
| | Options Pruned | 0/4 | 0/4 | 0/9 | 0/9 | 2/5 | 1/9 | 1/9 |
| | Data Use | 100.0% | 100.0% | 0.0% | 0.0% | 50.0% | 75.0% | 87.5% |

**Table 5:** Overview of performance, data use, and consent for all personalized models and systems on all datasets as measured by **test error**. We show the performance of models and systems built using **random forests**.

## D.3. Random Forests for Ranking (AUC)

| Dataset | Metrics | STATIC | | IMPUTED | | PARTICIPATORY | | |
|---|---|---|---|---|---|---|---|---|
| | | 1Hot | mHot | KNN-1Hot | KNN-mHot | Minimal | Flat | Seq |
| apnea $n = 1152, d = 26$ $\mathcal{G} = \{\text{age}, \text{sex}\}$ $|\mathcal{G}| = 6$ groups Ustun et al. [55] | Overall Performance | 0.825 | 0.824 | 0.822 | 0.806 | 0.823 | **0.944** | 0.942 |
| | Overall Gain | 0.008 | 0.006 | 0.004 | -0.012 | 0.005 | **0.126** | 0.124 |
| | Group Gains | $-0.004 - 0.009$ | $-0.005 - 0.012$ | $-0.004 - 0.009$ | $-0.005 - 0.012$ | $0.000 - 0.009$ | $0.058 - 0.157$ | $0.058 - 0.157$ |
| | Max Disparity | 0.012 | 0.017 | 0.012 | 0.017 | 0.009 | 0.098 | 0.098 |
| | Rat. Violations | **2** | **3** | **2** | **3** | 0 | 0 | 0 |
| | Imputation Risk | -0.004 | -0.005 | | | | | |
| | Options Pruned | 0/6 | 0/6 | 0/12 | 0/12 | 3/7 | 2/12 | 4/12 |
| | Data Use | 100.0% | 100.0% | 0.0% | 0.0% | 50.0% | 100.0% | 75.0% |
| cardio_eicu $n = 1341, d = 49$ $\mathcal{G} = \{\text{age}, \text{sex}, \text{race}\}$ $|\mathcal{G}| = 8$ groups Pollard et al. [43] | Overall Performance | 0.896 | 0.896 | 0.897 | 0.886 | 0.894 | **0.987** | 0.987 |
| | Overall Gain | 0.003 | 0.003 | 0.004 | -0.007 | 0.001 | **0.094** | 0.094 |
| | Group Gains | $-0.008 - 0.011$ | $-0.005 - 0.011$ | $-0.008 - 0.011$ | $-0.005 - 0.011$ | $0.000 - 0.004$ | $0.010 - 0.132$ | $0.010 - 0.130$ |
| | Max Disparity | 0.020 | 0.016 | 0.020 | 0.016 | 0.004 | 0.122 | 0.120 |
| | Rat. Violations | **3** | **4** | **3** | **4** | 0 | 0 | 0 |
| | Imputation Risk | -0.008 | -0.005 | | | | | |
| | Options Pruned | 0/8 | 0/8 | 0/27 | 0/27 | 7/9 | 10/27 | 10/27 |
| | Data Use | 100.0% | 100.0% | 0.0% | 0.0% | 12.5% | 100.0% | 87.5% |
| cardio_mimic $n = 5289, d = 49$ $\mathcal{G} = \{\text{age}, \text{sex}, \text{race}\}$ $|\mathcal{G}| = 8$ groups Johnson et al. [30] | Overall Performance | 0.884 | 0.883 | 0.884 | 0.881 | 0.885 | **0.955** | 0.954 |
| | Overall Gain | 0.000 | -0.001 | 0.001 | -0.002 | 0.001 | **0.071** | 0.071 |
| | Group Gains | $-0.005 - 0.006$ | $-0.006 - 0.013$ | $-0.005 - 0.006$ | $-0.006 - 0.013$ | $0.000 - 0.006$ | $0.016 - 0.108$ | $0.016 - 0.107$ |
| | Max Disparity | 0.011 | 0.019 | 0.011 | 0.019 | 0.006 | 0.092 | 0.090 |
| | Rat. Violations | **3** | **7** | **3** | **7** | 0 | 0 | 0 |
| | Imputation Risk | -0.005 | -0.006 | | | | | |
| | Options Pruned | 0/8 | 0/8 | 0/27 | 0/27 | 5/9 | 6/27 | 6/27 |
| | Data Use | 100.0% | 100.0% | 0.0% | 0.0% | 37.5% | 100.0% | 83.3% |
| coloncancer $n = 29211, d = 72$ $\mathcal{G} = \{\text{age}, \text{sex}\}$ $|\mathcal{G}| = 6$ groups Scosyrev et al. [45] | Overall Performance | 0.684 | 0.682 | 0.681 | 0.680 | 0.683 | **0.696** | 0.696 |
| | Overall Gain | 0.002 | 0.000 | -0.001 | -0.002 | 0.001 | **0.014** | 0.014 |
| | Group Gains | $-0.002 - 0.004$ | $-0.004 - 0.002$ | $-0.002 - 0.004$ | $-0.004 - 0.002$ | $0.000 - 0.004$ | $0.004 - 0.035$ | $0.004 - 0.031$ |
| | Max Disparity | 0.006 | 0.007 | 0.006 | 0.007 | 0.004 | 0.030 | 0.026 |
| | Rat. Violations | 0 | 0 | 0 | 0 | 0 | 0 | 0 |
| | Imputation Risk | -0.002 | -0.004 | | | | | |
| | Options Pruned | 0/6 | 0/6 | 0/12 | 0/12 | 3/7 | 2/12 | 5/12 |
| | Data Use | 100.0% | 100.0% | 0.0% | 0.0% | 50.0% | 100.0% | 75.0% |
| lungcancer $n = 120641, d = 84$ $\mathcal{G} = \{\text{age}, \text{sex}\}$ $|\mathcal{G}| = 6$ groups Scosyrev et al. [45] | Overall Performance | 0.849 | 0.849 | 0.848 | 0.849 | 0.848 | **0.856** | **0.856** |
| | Overall Gain | 0.002 | 0.001 | 0.001 | 0.001 | 0.000 | **0.008** | **0.008** |
| | Group Gains | $-0.001 - 0.003$ | $-0.001 - 0.002$ | $-0.001 - 0.003$ | $-0.001 - 0.002$ | $0.000 - 0.003$ | $0.002 - 0.020$ | $0.002 - 0.020$ |
| | Max Disparity | 0.004 | 0.003 | 0.004 | 0.003 | 0.003 | 0.018 | 0.018 |
| | Rat. Violations | **1** | **1** | **1** | **1** | 0 | 0 | 0 |
| | Imputation Risk | -0.001 | -0.001 | | | | | |
| | Options Pruned | 0/6 | 0/6 | 0/12 | 0/12 | 2/7 | 1/12 | 2/12 |
| | Data Use | 100.0% | 100.0% | 0.0% | 0.0% | 66.7% | 100.0% | 91.7% |
| saps $n = 7797, d = 36$ $\mathcal{G} = \{\text{HIV}, \text{age}\}$ $|\mathcal{G}| = 4$ groups Allyn et al. [3] | Overall Performance | 0.921 | 0.922 | 0.922 | 0.906 | 0.921 | **0.966** | **0.966** |
| | Overall Gain | 0.003 | 0.004 | 0.003 | -0.012 | 0.002 | **0.048** | **0.048** |
| | Group Gains | $-0.002 - 0.010$ | $-0.002 - 0.013$ | $-0.002 - 0.010$ | $-0.002 - 0.013$ | $0.000 - 0.010$ | $0.009 - 0.109$ | $0.009 - 0.109$ |
| | Max Disparity | 0.012 | 0.015 | 0.012 | 0.015 | 0.010 | 0.100 | 0.100 |
| | Rat. Violations | **2** | **2** | **2** | **2** | 0 | 0 | 0 |
| | Imputation Risk | -0.002 | -0.002 | | | | | |
| | Options Pruned | 0/4 | 0/4 | 0/9 | 0/9 | 2/5 | 2/9 | 2/9 |
| | Data Use | 100.0% | 100.0% | 0.0% | 0.0% | 50.0% | 100.0% | 87.5% |

**Table 6:** Overview of performance, data use, and consent for all personalized models and systems on all datasets as measured by **test auc**. We show the performance of models and systems built using **random forests**.

