# OpenReview forum: "Participatory Personalization in Classification"
_ICML.cc/2023/Workshop/IMLH — IMLH 2023 Poster_

### Official Review · Reviewer_Evbr · 2023-06-17
**Review for Submission 24**

**Rating:** 9
**Confidence:** 3

**Review:**

This paper introduces participatory systems, which enable individuals to report personal information at prediction time, inform consent, and lead to significant performance improvements across reporting groups.

Pros:
1. This introduced family of classification models called participatory systems allow individuals to report personal information at prediction time and inform consent by showing how reporting personal information will impact predictions.
2. The proposed idea helps to address the ethical concerns surrounding personal data use by emphasizing the importance of informed consent and responsible data collection.
3. While the empirical study focuses on clinical prediction tasks, the concept of participatory systems has the potential to be applied to other domains, expanding its impact and benefits.
4. The paper provides insights into the development and implementation of participatory systems, offering a practical approach for incorporating informed consent into classification models.

Cons:
1. Implementing participatory systems in real-world scenarios may pose challenges, such as designing user interfaces for data reporting, ensuring data accuracy, and managing the trade-off between performance gains and data privacy.
2. The paper does not address potential barriers to user adoption of participatory systems, such as user reluctance to report personal information or concerns about data misuse, which may impact the practicality and effectiveness of the approach.

---

### Official Review · Reviewer_gcfZ · 2023-06-19
**This article proposes a novel focus, but I am not very clear about the core parts of it.**

**Rating:** 6
**Confidence:** 1

**Review:**

The paper is very well written, and highlights the issue of machine learning models using personal information without consent and the introduction of participatory systems. These systems allow individuals to opt into personalization at the time of prediction, improving performance and data utilization. Through empirical studies, it is shown that participatory systems facilitate and inform consent for all groups reporting personal data.
Concerns arise regarding individuals opting out of personalization, hindering data collection for future improvements. The principle of purpose specification guides data collection, where individuals can voluntarily report for monitoring or model enhancement. Successful personalization depends on individual disclosure preferences, necessitating tailored approaches to communicate benefits and risks.

---

### Official Review · Reviewer_gANm · 2023-06-19
**The paper deals with an interesting problem and proposes a feasible framework, but there is concern about the framework design and the evaluation**

**Rating:** 6
**Confidence:** 4

**Review:**

The paper proposes a novel participatory systems, a family of classification models that allow individuals to opt into personalization at prediction time, providing consent and benefiting from personalized predictions.

The idea of personalize model is novel and provides a feasible solution to the problem. The largest concern is that there is a lack of adequate baseline methods for comparison in the evaluation section, which makes the claims of the proposed framework to be doubtable.

In the real-world applications, the primary goal is still to get the best prediction performance. It would be necessary for the paper to report what is the truly best performance of each dataset if participants report all information. The current results seemingly claim that the proposed framework to use a subset of features have the best performance for all of the dataset. However, It is unrealistic that for all the datasets, the chosen group information all happen to be the one that "worsens" the prediction. Even if the "worsenlisation" effects happen to be severe in all of the datasets, there are various method on market to avoid the so-called "worsenlisation" effect of the data, then they are important methods needed to be compared. To make a valid evaluation, it is of great importance to report the performance of other competitors without feature selection.

---

### Meta-Review · Area_Chair_JzoG · 2023-06-20

**Recommendation:** Accept (Poster)
**Confidence:** 3

**Metareview:**

The work proposes a model-agnostic algorithm and an empirical study to address the timely problem of participatory personalization. Given the novelty and validity of the work, I recommend it to IMLH23 and suggest the authors incorporate the reviewers' comments in their revisions.

---

### Decision · Program_Chairs · 2023-06-20

Accept (Poster)